# Bioinformatic Approach of B and T Cell Epitopes of PLD and CP40 Proteins of *Corynebacterium pseudotuberculosis ovis* Mexican Isolate 2J-L towards a Peptide-Based Vaccine

**DOI:** 10.3390/ijms25010270

**Published:** 2023-12-23

**Authors:** Maria Carla Rodríguez-Domínguez, Roberto Montes-de-Oca-Jiménez, Juan Carlos Vázquez-Chagoyán, Pilar Eliana Rivadeneira-Barreiro, Pablo Cleomenes Zambrano-Rodríguez, Martha Elba Ruiz-Riva-Palacio, Adriana del Carmen Gutiérrez-Castillo, Siomar de-Castro-Soares, Patricia Vieyra-Reyes, Gabriel Arteaga-Troncoso

**Affiliations:** 1Research and Advanced Studies in Animal Health Center, Faculty of Veterinary Medicine and Zootechnics, Autonomy University of the State of Mexico, Km 15.5 Toluca Pan-American Highway Atlacomulco, Toluca C.P. 50200, State of Mexico, Mexico; mariacarlarodriguezdominquez@gmail.com (M.C.R.-D.); jcvch@yahoo.com (J.C.V.-C.); acgutierrezc@uaemex.mx (A.d.C.G.-C.); pvieyrar@uaemex.mx (P.V.-R.); 2Sor Juana Inés de la Cruz School, Autonomy University of the State of Mexico- AMECAMECA, Amecameca de Juarez C.P. 56900, State of Mexico, Mexico; prometeoruiz@hotmail.com; 3Department of Veterinary Medicine, Faculty of Veterinary Sciences, Technical University of Manabí, Urbina Avenue, Portoviejo C.P. 130105, Portoviejo, Ecuador; pilar.rb26@hotmail.com (P.E.R.-B.); drpablozambrano@gmail.com (P.C.Z.-R.); 4Department of Microbiology, Immunology and Parasitology, Federal University of Triângulo Mineiro, Av. Frei Paulino, 30-Nossa Sra. da Abadia, Uberaba C.P. 38025-180, Minas Gerais, Brazil; siomars@gmail.com; 5Department of Cellular Biology and Development, National Institute of Perinatology, Lomas de Chapultepec IV Secc, Miguel Hidalgo, Mexico City C.P. 11000, Mexico; drgarteagat@yahoo.com.mx; 6Military School of Health Officers, University of the Mexican Army and Air Force, SEDENA, Mexico City C.P. 11650, Mexico

**Keywords:** *Corynebacterium pseudotuberculosis ovis*, T cell epitopes, B cell epitopes, PLD, CP40, peptides, caseous lymphadenitis

## Abstract

Mapping B and T cell epitopes constitutes an important action for peptide vaccine design. PLD and CP40 virulence factors of *Corynebacterium pseudotuberculosis* biovar *ovis*, a causal agent of Caseous Lymphadenitis, have been evaluated in a murine model as good candidates for vaccine development. Therefore, the goal of this work was to in silico analyze B and T cell epitopes of the PLD and CP40 proteins of a Mexican isolate of *Corynebacterium pseudotuberculosis ovis*. The Immune Epitope Data Base and Resource website was employed to predict the linear and conformational B-cell, T CD4+, and T CD8+ epitopes of PLD and CP40 proteins of *Corynebacterium pseudotuberculosis ovis* Mexican strain 2J-L. Fifty B cell epitopes for PLD 2J-L and forty-seven for CP40 2J-L were estimated. In addition, T CD4+ and CD8+ cell epitopes were predicted for PLD 2J-L (MHC I:16 epitopes, MHC II:10 epitopes) and CP40 2J-L (MHC I: 15 epitopes, MHC II: 13 epitopes). This study provides epitopes, paying particular attention to sequences selected by different predictor programs and overlap sequences as B and T cell epitopes. PLD 2J-L and CP40 2J-L protein epitopes may aid in the design of a promising peptide-based vaccine against Caseous Lymphadenitis in Mexico.

## 1. Introduction

Reverse vaccinology is a branch of immunoinformatics, and is also known as the in silico search for vaccine candidates. The computational analysis constitutes an alternative tool, allowing the characterization of potential vaccine candidates, avoiding pathogen microorganism cultures, the use of experimental animals in the early stages of design, and achieving significant savings in research time and costs [1]. This technology was first used in the prediction of antigens for the development of a vaccine against Meningococcus serogroup B [2].

Bioinformatics programs allow the prediction based on different antigen characteristics such as structure, adherence, fusion, antigenicity, binding to Major Histocompatibility Complex (MHC) classes I and II, binding sequences to proteasomes, and B and T lymphocyte activators [3]. 

In silico programs to determine epitopes, specific regions of a protein that can be recognized by antibodies (B cell epitopes), or amino acid sequences that can interact with TCRs in the context of MHC I or MHC II-mediated antigen presentation (T cell epitopes) are considered of great practical interest for vaccine design [4].

Immune Epitope Database and Analysis Resource (IEDB) is a website to allow B and T cell epitope predictions, providing a regularly updated compilation of binding epitopes and their affinities. It constitutes a prediction tool accessible through a single intuitive web interface that incorporates different programs and algorithms for the general characterization of a protein [5].

In addition, B cell epitope mapping is a promising approach to identifying the main antigenic determinants of a protein, linear epitopes for continuous amino acid residues next to each other in the primary structure or discontinuous epitopes, also known as conformational ones, based on residues close to each other in the three-dimensional structure of the protein [6].

In fact, B cell and T cell epitope prediction allows the formulation of vaccines based on peptides to substitute the use of the entire protein with the guarantee of producing specific antibodies or a cellular immune response [3], which represents an alternative to avoid the use of toxigenic nature antigens. 

Phospholipase D (PLD) [7] and Endoglycosidase CP40 [8] are virulence factors of *Corynebacterium pseudotuberculosis* biovar *ovis*, a gram-positive rod that cause Caseous lymphadenitis (CLA), a source of economic loss to the sheep and goat industries. CLA characteristic lesions are associated with encapsulated cutaneous and visceral abscesses with pyogranulomas formation, a response of the immune system to contain the infections in the host’s tissues, but which makes it difficult to eliminate the bacteria [9,10]. 

Conventional antibiotic treatment and commercial vaccines are not efficient at controlling the disease [11]. The fact that *C. pseudotuberculosis ovis* is an intracellular facultative bacterium makes vaccines’ cellular immune response induction an important role for pathogen elimination [12]. 

Most vaccine formulations from CLA include the PLD toxin [13], because it is considered the principal virulent factor of *C. pseudotuberculosis ovis*, especially for contributing to bacterial spread in the host and their enzymatic capacity to hydrolyze the sphingomyelin of the cell membrane, causing direct damage to endothelial cells and dermo necrosis [14]. 

Antibodies against PLD are generated to neutralize the toxin effect, but the intracellular condition of the bacteria makes it necessary to direct the action of the immune system to obtain cell-mediated immunity, which is more difficult using conventional vaccine strategies. Antibodies might help to protect animals against infection, but full protection by any vaccine model for CLA must provide better stimulation of cellular immunity, such as the activation of T CD8+ cells and IFN-γ secretion [12,13].

In that way, virulence factor CP40, a protein with enzymatic activity that can hydrolyze Fc regions of ovine antibodies [8], has been previously identified as a protective antigen against CLA [15,16]. Vaccine formulations with CP40 induce a cellular immune response, with the highest survival percentage in animal challenge studies [17]. PLD [18,19] and CP40 [17,20] immunogens’ potential has been evaluated in some experimental assays, and they are capable of enhancing humoral and cellular immune responses. 

Mexico has a sheep and goat industry that is still developing, so it is important to study the disease to improve control measures for future eradication. Previous studies [21,22] have shown the presence of this pathogenic agent in Mexican small ruminant herds, which, added to the fact that no vaccines are commercialized in the country, arouses interest in solving this problem. 

Previously, the complete sequence of the *pld* (GenBank accession number: OL347711) and *cp40* (GenBank accession number: OL347712) genes of *Corynebacterium pseudotuberculosis ovis* isolate 2J-L, obtained from Jalisco State, Mexico, was reported [23]. Since those virulence factors have been studied as promising vaccine antigens, this work aimed to in silico identify B and T cell epitopes of PLD 2J-L and CP40 2J-L to determine possible peptides for a vaccine design that combines the specific components that allow effective immune system activation.

## 2. Results

### 2.1. B lymphocyte Epitope Predictions for PLD 2J-L and CP40 2J-L Proteins

B lymphocyte epitopes were predicted independently for PLD 2J-L and CP40 2J-L proteins, and results were compared in terms of their different inmunogenic characteristics. A total of 10 B-cell linear epitopes for PLD 2J-L and 12 B-cell linear epitopes for CP40 2J-L of ≥5 amino acids length were identified by BepiPred 2.0 software (Table 1). 

B cell epitope predictions and analysis were also performed using the IEDB website. Results show B cell epitopes determined by different immunogenic properties of the proteins, such as antibody binding, surface exposure, and antigenicity. A range of peptides composed of 5 to 35 amino acid residues were analyzed and, taking into account the cut-off value established for each criterion, the sequences that make up the B cell epitopes for PLD 2J-L and CP40r 2J-L were selected. Peptides were mapped to the PLD 2J-L and CP40 2J-L sequences, pointing out which B cell epitopes are most likely to interact with antibodies (Abs), to be on the surface, and to be most antigenic. In addition, sequences that share the properties of Ab binding and surface exposure were identified (Figure 1).

Linear (Table 2) and conformational (Table 3) epitopes of B cells of the PLD 2J-L and CP40 2J-L proteins were estimated using the ElliPro program, taking into account the PDB format of the three-dimensional structure of the proteins.

A total of 13 linear epitopes were identified for each protein, and scores above the cut-off value were taken into account for peptide selection. Epitopes were identified with a minimum size of 4 and 5 residues and a maximum size of 33 and 46 for PLD 2J-L and CP40 2J-L, respectively. The number 1 epitope of both proteins presented the highest score value, 0.856 for PLD 2J-L and 0.807 for CP40 2J-L.

A schematic representation of B-cell linear epitopes for PLD 2J-L and CP40 2J-L was performed using the JMol structure visualization program (Figure 2).

For predicting B-cell conformational epitopes, a protein structure PDB format is re-quired. That is why the structure of PLD 2J-L and CP40 2J-L proteins was performed using the Phyre2 web portal, which produces potential 3D models of the proteins, based on the alignment of sequences and crystal structures of known proteins. PLD 2J-L and CP40 2J-L amino acid sequences were used to compare against an updated library of protein sequences, using PSI-BLAST, with the detection of 1000 homologues for each protein and PBD format archives generated and used for structural epitope prediction. 

B lymphocyte conformational epitopes, composed of non-contiguous amino acid residues in the primary sequence of the protein but joined by a folding structure, were predicted based on the 3D model of the PLD 2J-L and CP40 2J-L proteins. The analysis of the structures allowed for the identification of eight epitopes for PLD 2J-L and three for CP40 2J-L (Table 3). The largest epitopes of the PLD 2J-L protein are 1 and 7, with scores of 0.822 and 0.609, respectively. The CP40 2J-L epitope 2 was estimated with a value of 0.683 spanning 171 residues of the protein, which includes a large part of the molecule as a possible B-cell binding conformational epitope.

In addition, a schematic representation of discontinuous B cell epitope arrangement was performed for both proteins, allowing spatial visualization of antigenic epitopes (Figure 3).

Fifty B lymphocyte epitopes for PLD 2J-L and forty-seven for CP40 2J-L (9-12 by BepiPred 2.0, 9-8 by Emini Surface Accessibility Scale, 11-11 by Antigenicity Kolaskar—Tongaonkar scale, and 13-13 linear and 8-3 discontinuous by ElliPro using protein structures in PDB format) were estimated using bioinformatics tools. The resulting data set represents the B cell epitopes for PLD 2J-L and CP40 2J-L estimated by different predictors. Epitope prediction by different programs is not identical, so regions that overlap in each epitope prediction were identified (Table 4).

In this work, the use of various predicted programs allowed the identification of regions in the protein sequence as epitopes, with variation in length and the presence or absence of certain amino acids, but most likely considering an epitope. 

Comparing the different epitopes identified by the predictors, we were able to establish 11 regions in the PLD 2J-L and 12 in the CP40 2J-L sequences, which were predicted by at least two different criteria: PLD 2J-L (24AAPVVHNP31, 79DGIPTSAGATAE90, 117DYCRD121, 133DLARKYL139, 153TVGGP157, 179QDVLND184, 185FARSENKILTKQKI198, 211GNCYGTWNRT220, 229EARDQGKL236, 245ATGQDA250, 276HADTE280). CP40 2J-L (5PRSVSRLITVGITSALFASTFSAVASA31, 33SATLSKEPLKAS44, 50DTVGVQTTCN58, 70DKAIQLKDDDPWKDKLQVKLTD93, 145LNKIK149, 158VEDDYKYRE166, 193VEKQLNLK200, 221NEGKKPDHE229, 240DNAQ243, 294EENDTNRFLTAVGEVNKSG312, 337DGRTYD342, 366 GESSTDLGKPT376).

### 2.2. T Lymphocyte Epitope Predictions for PLD 2J-L and CP40 2J-L Proteins

The cellular immune response is extremely important, especially the activation of T lymphocytes, for the resolution of intracellular bacterial infections. Since T cell epitopes are bound in a linear form to MHCs, the capacity of PLD 2J-L and CP40 2J-L proteins to activate T cells was evaluated in the context of antigen presentation by MHC classes I (Table 5) and II (Table 6). In addition, the amino acid residues that were identified as part of some B cell epitopes are also shown.

The PLD 2J-L protein sequences with the greatest potential to activate T CD8+ cells in the context of different MHC I alleles: position 56 VAIGANAL 63 (alleles H-2-Kb and H-2-Db), 139 LEPAGVRV 146 (alleles H-2-Kd, H-2-KK, and H-2-Qa1) and sequence 202 GYYNI 206 (alleles H-2-Kd, H-2 Kb, and H-2-KK). For the CP40 2J-L protein, the sequences that were recognized by more than one type of allele were sequences 137 RTVGAQL 143 (alleles H-2-Qa1 and H-2-Kb) and 205 KIMGAFSEL 213 (alleles H-2-Kb and H-2-Qa1). Nevertheless, none of the proteins presented epitopes with the capacity to be presented by MHC class I alleles H-2-Dd, H-2-Ld, or H-2-Qa2. Also, results show sequences that were selected as B cell epitopes (Table 5). 

Epitopes predicted for presentation on T CD4+ in the context of MHC II for both proteins were not capable of being recognized by more than one type of allele. Neither of the two proteins present epitopes for T cell activation in the context of the MHC II allele H2-Ied. 

However, the analysis made it clear which T epitopes are also considered epitopes of B cells (Table 6). Sequence 24′VHNPASTAN′38 of PLD 2J-L could activate T cells (MHC II-H2-IAb Allele) and B cells as part of epitopes predicted by BepiPred 2.0, Emini Surface scale, Antigenicity scale, and Ellipro programs. Following the same analysis for CP40 2J-L, the sequence 286′YAHPEENDT′300 is considered a T cell epitope (MHC II-H2IAb Allele) and a B cell epitope according to all the programs used.

## 3. Discussion

### 3.1. B Lymphocyte Epitopes Prediction for PLD 2J-L and CP40 2J-L

The results showed that PLD 2J-L and CP40 2J-L are multi-epitopic proteins with the potential as candidates for the development of vaccines and diagnostic tools for Caseous lymphadenitis. Several of the predicted B cell epitopes for PLD 2J-L and CP40 2J-L were classified as exposed epitopes, located on the surface of the respective proteins. These regions can be used as synthetic peptides to stimulate an antibody response, ensuring better targeting of the immune system during a natural infection. 

Some of the sequences were identified as antigenic and antibody-binding epitopes, all with high conservation and predicted by BepiPred 2.0 and the Antigenicity Kolaskar–Tongaonkar Scale.

Almost the entire sequence of the PLD 2J-L and CP40 2J-L proteins was mapped and included as a B cell epitope, demonstrating their immunogenic potential. Most of the complete set of predicted linear epitopes was included in some conformational epitopes, which can be used as a basis for protein–protein interaction studies using three-dimensional structures [24].

B cell epitope number 2 of PLD 2J-L is formed by the sequence of the catalytic loop, which is a conserved region important for the enzymatic activity of the protein. In addition, we found that the epitopes at position (51–100) of PLD 2J-L, predicted by BepiPred 2.0 and ElliPro, include a highly conserved sequence of the catalytic loop, also forming part of a conformational epitope. Since this region is conserved between PLDs of different strains of *C. pseudotuberculosis*, the peptides that include it could be a good candidate for vaccine development. In addition, the toxin PLD is the main virulence factor of the bacteria, and usually, vaccine strategies use the complete protein without first considering the inactivation of their biological activity. Toxin inactivation could negatively impact PLD protective potential, a fact that can be exemplified by Hodsong and colleague’s experiments, where an attenuated strain called Toxminus was used for vaccine development, and PLD active site modification decreased the protective capacity of the vaccine [7,25,26,27]. More recently, genetic modification for the recombinant PLD active site also did not allow for reaching a survival percentage greater than 57% in vaccinated and challenged animals [28]. Other studies have used recombinant PLD as a vaccine antigen evaluated in murine models, presenting different protective percentages of 30% [18], 40% [29], and 42.86% [28] after challenge with virulence strain MIC-6 (10^4^ CFU), with the common denominator being the inactivation of the toxin’s biological activity with formaldehyde. We could say that peptide 2 of PLD 2J-L, which involves the protein active site, could be a determinant for enhanced protection. PLD anti-exotoxin antibodies protect against tissue damage and bacterial dissemination, which makes it very important to include this antigen in vaccine formulation.

Also, results shown in Table 3 indicate that the region of the PLD protein made up of amino acid residues 50 to 240 has the greatest potential since more epitopes were identified in this region estimated by the different programs.

The immunological potential of CP40 has been previously studied in silico, with the description of six immunodominant epitopes and hydrophobic areas of the protein that could interact with the Toll-like 2 receptor [30]. These sequences are conserved in CP40 2J-L, providing evidence for the ability of this protein to bind TLR2. Toll-like receptors (TLR) are involved in the innate and adaptive responses of the immune system [31]. 

Recently, six peptides from the CP40 protein were evaluated for vaccine formulation. We could identify those peptides as being part of B cell epitopes predicted in our work by the BepiPred 2.0 program. Peptide 1 was included in epitope number 5, peptides 2 and 4 are included in epitope number 8, and peptide 3 is part of epitope number 3. For peptide 5, we could identify the sequence as part of an epitope classified as antigenic and surface-exposed. In addition, peptide 6 was classified as a TLR2 activator. Each peptide was evaluated independently in vaccine formulations and a murine model. Different protective values were obtained; for example, peptides 4 and 6 provided 10% protection, while peptide 5 provided 20%. In addition, peptides 1 and 2 provided 30% and 40%, respectively, the highest protection values. Although all groups were affected during the challenge with a virulence strain, they also showed the production of total IgG antibodies and cytokines (IL-2, IL-4, IL-6, IFN-γ, and TNF-α), indicating a possible activation of the Th1 type response. The authors suggest the use of adjuvants based on PAMPs to improve the immune response offered by these peptides [32]. Since peptide sequences are also included on some predicted epitopes in our work, we consider it interesting to evaluate a multi-epitope vaccine with the combination of all peptides in the same formulation to enhance levels of protection. 

Previously, no epitope mapping has been performed for these proteins, comparing the results obtained by predictors with different selection criteria such as physicochemical properties, surface epitope exposure, antigenic properties, and three-dimensional structure. In addition, computational prediction tools based on antigen structures constitute an alternative for in silico epitope prediction [33,34]. However, structural information is only available for a few antigens, and in the vast majority of cases, one is left with analyzing the primary sequence. Since there is no crystal structure for PLD or CP40, we used the Phyre 2.0 program to predict the PDB format for both proteins.

Our work presents for the first time a summary of the overlapping sequences that were predicted by different criteria and included as B epitopes by more than two predictors. The results in Table 3 provide details of the protein sequences and regions most likely responsible for the humoral response. For peptide selection for vaccine development, we recommended the epitopes that were identified by more than one criterion and have a high score value such as 9 MREKVVLFLSIIMAIMLPVGNAAAAPVVHNPAS 33 of PLD and 9 MHNSPRSVSRLITVGITSALFASTFSAVSASATLSKEPLKASPG 46 of CP40. 

### 3.2. T Lymphocyte Epitope Prediction of PLD 2J-L and CP40 2J-L

The intracellular life of *C. pseudotuberculosis* biovar *ovis* allows it to survive and multiply within macrophages, so the induction of a cellular immune response is crucial to achieving an effective vaccine that generates complete protection [35]. T CD8+ cells are responsible for eliminating intracellular pathogens [36]. Therefore, new studies are needed to understand how to modulate this immune response. 

MHC class I is expressed in all cells except erythrocytes, neurons, and the fetal trophoblast. Peptides presented by this class of MHC are smaller, between 8 and 10 amino acids. MHC I only presents antigens to the TCR (T cell receptor) of cytotoxic T lymphocytes, cells with CD8+ molecules that specifically interact with MHC I [37,38]. 

The MCH class II can present peptides between 13 and 17 amino acids. MHC II is only possessed by a very particular group of cells of the immune system, which have been called professional antigen-presenting cells (APCs), and among them are B lymphocytes, monocytes, dendritic cells, epithelial cells of the thymus, Langerhans cells, and cells from Kupffer. These molecules only present antigens to T helper lymphocyte receptors because on their surface there is molecule CD4 that binds to one of the MHC II chains during antigen presentation [38].

MHC genes are highly polymorphic, which means that there are many different alleles in the individuals within a population. Polymorphism is so high that in a mixed population, no two individuals have the same set of genes and MHC molecules. The polymorphic regions in each allele are located in the peptide contact region, which is to be displayed to the lymphocyte. For that reason, the region of contact for each allele of the MHC molecule is highly variable, as polymorphic MHC residues will create specific clefts that only certain types of peptide residues can enter. This imposes a very specific binding between the MHC molecule and the peptide and implies that each MHC variant will be able to specifically bind only those peptides that fit properly into the MHC molecule’s cleft, which is variable for each allele [39,40].

Numerous T cell epitope-mapping algorithms have been established and used to rapidly identify putative T cell epitopes [41], mostly directed at the detection of the epitopes for human alleles. We used the IEDB analysis resource database that uses NetMHCpan as a prediction method [42] to generate a high quantitative prediction. Even when this tool constitutes one of the few databases that include a variety of MHC alleles from different organisms, such as humans, chimpanzees, macaques, gorillas, cows, pigs, and mice, there is still limited success for prediction of other species like sheep, caprine, or equine, due to an insufficient training data set.

Since the murine model has been used for experimental CLA vaccine evaluation, constituting a good model for abscess formation and systemic infection [13], we used mouse allele information to search T cell epitopes in the context of presentation by MHC-I and MHC-II. 

Analysis revealed peptides from PLD 2J-L and CP40 2J-L that may interact with a broad range of alleles from MHC class I (H-2-Kb, H-2-Kk, H-2Kd, H-2-Db, H-2Qa1) and MHC class II (H2-IAb, H2-IAd).

The selection of regions that share B and T cell activation capacity would be the most opportune for the development of a peptide vaccine, since they would guarantee the stimulation of the humoral and cellular responses.

This study was carried out with MHC alleles as references since the site for prediction does not have any information for ovine or caprine MHC molecules. The genes that code for the MHC in sheep are located on chromosome 20, initially called OLA based on the HLA nomenclature for humans, but later the name Ovar-MHC was unofficially established. In sheep, the class I region is poorly characterized, and different studies have proposed that there are four loci: OLA-A, B, C, and D. Ovine MHC- class I genes are more similar to bovine than to murine class I genes [43]; however, the IEDB database also does not contain information on bovine alleles, so we used the murine allele data. Like MHC class I molecules from other species, those from sheep were found to be distributed in lymphocytes and non-lymphoid tissues [44]. The high degree of polymorphism in the genes that encode the MHC impacts the individual immune response during vaccination; for this reason, different alleles were evaluated in the presentation of the T cell epitopes.

The results obtained show that peptides of PLD 2J-L and CP40 2J-L are capable of being presented by MHC I and MHC II alleles in the murine model. The inclusion of these epitopes in a vaccine formulation could produce adequate stimulation of the specific immune response, although it must be taken into consideration that these results were obtained in a murine model. 

Cytotoxic T lymphocytes (CTLs) recognize antigens presented by the MHC class I receptor found on all nucleated cells. Effector CTL cells have the ability to recognize and kill cells infected with the pathogen, constituting an important component of protective immunity. However, it has been proven that during chronic infections T cells may suffer from functional deterioration, which is one of the main reasons for the inability of the host to eliminate the persistent pathogen. T cells that are generated during the early stages of infection gradually lose function during the course of chronic infection. This phenomenon of exhausted T lymphocyte production has been characterized in mice and humans with established chronic infections by viral agents [45]. For these reasons, it is essential to activate memory T cells. Despite some similarities at the molecular, epigenetic, metabolic, and functional levels between effector CD8+ T cells and memory CD8+ T cells, the latter persist for long periods of time and are capable of proliferating after re-encountering the antigen [36]. The results achieved in this work correspond to the prediction of epitopes capable of activating the global population of CD8+ T cells; therefore, experimental studies will be required to specifically evaluate the subset of memory CD8+ T cells [36].

There is experimental evidence indicating that formulations with recombinant CP40 protein were more efficient in providing protection associated with a Th1 cell response in the absence of a Th2 profile [20]. The identification of a significant increase in the levels of the IgG2 isotype is related to a Th1 cellular response, cells mainly involved in immunity against intracellular pathogens, activation of macrophages and cytotoxic T cells, opsonization production, and complement activation [17].

The role of TLR2 in the homeostasis of memory T CD8+ cells is related to the direct control of the proliferation of these cells and the secretion of IFN-ɣ. A study demonstrated the participation of TLR2 in the activation of memory T CD8+ cells, during an established chronic viral infection in a murine model, causing an increase in the proliferation and expansion of these cells, induced by IL-7 both in vitro and in vivo [46,47]. Another study found that TLR2 increases the activation of T CD8+ cells and allows the generation of functional memory cells in response to low or inefficient TCR signals [48]. Taken together, these data support the idea that CP40 2J-L peptides that induce TLR2 signaling could directly contribute to the maintenance of CD8+ T cell memory.

T CD4+ cells activate other cells of the immune system, such as B cells, CD8+ T cells, and macrophages. Under conditions of chronic infection, in which CD8+ T cells take several months or more to clear the infection, CD4+ T cells play an important role in maintaining CD8+ T cell activity [49]. Experimental studies have considered the induction of a Th1 response (CD4+ T cells) to be relevant for the activation of CD8+ T cells in the context of MHC I, thus achieving an efficient response against this pathogen [18,19].

## 4. Materials and Methods

### 4.1. PLD 2J-L and CP40 2J-L Proteins Sequences and Structure 

The published PLD 2J-L (accession: UYI58181) and CP40 2J-L (accession: UYI58182) protein sequences were retrieved from the National Center for Biotechnology Information (NCBI) http://www.ncbi.nlm.nih.gov/ (Accessed on 6 March 2023). Proteins belong to isolate 2J-L of *Corynebacterium pseudotuberculosis* biovar *ovis* obtained from a clinical case of Caseous Lymphadenitis in a sheep from the State of Jalisco, Mexico [23]. The Phyre2 web portal was used for modeling protein structures and producing structures in PDB format, which is necessary for conformational epitope prediction [50]. This web portal includes different programs such as HHpred 1.51 for template detection [51], Psi-pred 2.5 for secondary structure prediction [52], Disopred 2.4 for structure disorder prediction [53], Memsat SVM for transmembrane domain prediction [54], multi-template modeling, Poing 1.0 ab initio for template less structure modeling [55], and the the Jmol viewer for a 3D view of structures [56].

### 4.2. Linear and Conformational B Cell Epitope Prediction Tools 

B cell epitope prediction was carried out using available tools in the Immune epitope database and resource website. Linear epitopes were predicted with BepiPred 2.0 (score value: 0.5) [57], the Emini surface scale for surface accessibility prediction (score value: 1.0) [58], and the Kolaskar and Tongaonkar antigenicity scale (score value: 0.5) [59]. Ellipro http://tools.iedb.org/ellipro/ (Accessed on 5 June 2023) [33] was used for linear and conformational epitopes prediction (score value: 0.5) using 3D protein structures in PDB format. B cell epitopes predicted by Ellipro were visualized in the Jmol program. Manual analyses were done for the identification of sequences predicted by more than one predictor.

### 4.3. T Cell Epitope Prediction

T CD8+ epitope prediction was done against MHC class I alleles (H-2- Kb, H-2-Kd, H-2-Kk, H-2-Db, H-2-Dd, H-2-Ld, H-2-Qa1, H-2-Qa2)] using the NetMHCpan EL 4.1 methods [60] from the IEDB website. The epitopes with a score rank ≤0.5 were shortlisted for further analysis. In parallel, T CD4+ epitope prediction was done using NetMHCII-pan 4.1 EL methods [61] against the MHC class II alleles (H2-IAb, H2-IAd, and H2-Ied). The epitopes (15-mer) with a score rank ≤0.5 were shortlisted. Sequences that appeared as part of some B cell epitopes were highlighted.

## 5. Conclusions

The prediction of B and T cell epitopes in an antigen constitutes one of the important steps in the design of multi-epitope-based vaccines. B cell epitopes were predicted independently using selected prediction tools, and the results were compared to obtain conserved sequences by the different programs. T CD8+ and CD4+ cell epitopes and their corresponding MHC restricted alleles were identified by prediction tools provided by the immune epitope database (IEDB). Results confirmed the potential inmunodominant epitopes of PLD and CP40, containing ‘promiscuous’ or ‘universal’ epitopes that cover diverse MHC haplotypes. This knowledge can then be utilized to improve peptide-based vaccine design for Caseous Lymphadenitis.

## Figures and Tables

**Figure 1 ijms-25-00270-f001:**
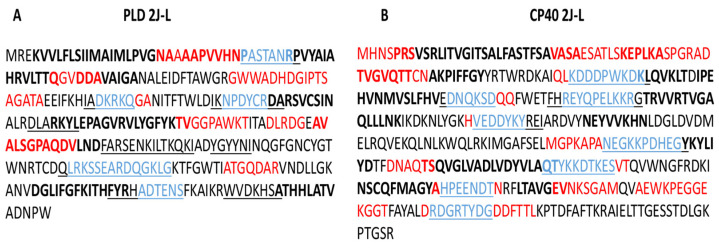
B cell epitopes of PLD 2J-L (**A**) and CP40 2J-L (**B**) using the IEDB website. Epitopes most likely to interact with antibodies are colored red, surface epitopes are indicated by an underlined sequence, antigenic epitopes are indicated in bold letters, and the sequence that shares the ability to bind antibodies and is further exposed to the surface is shown in blue.

**Figure 2 ijms-25-00270-f002:**
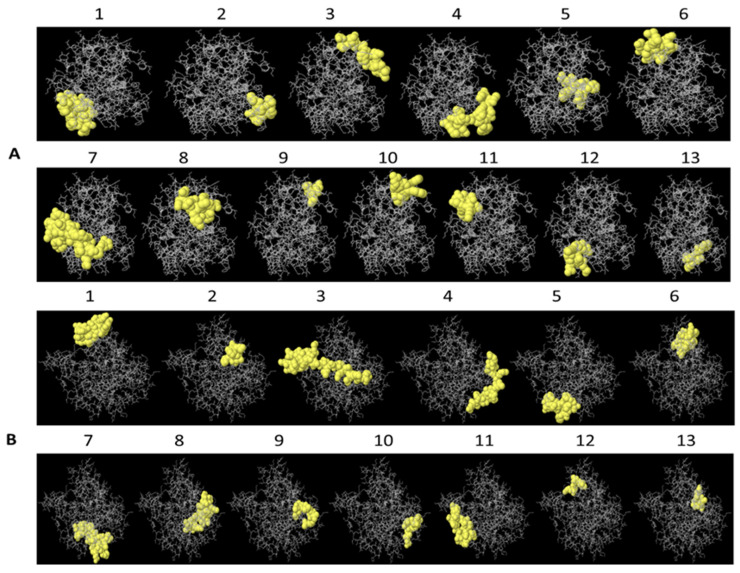
A schematic display of B lymphocyte linear epitopes of PLD 2J-L (**A**) and CP40 2J-L (**B**) estimated by ElliPro, based on the 3D structure of the proteins. The numbers represent the epitopes of Table 2.

**Figure 3 ijms-25-00270-f003:**
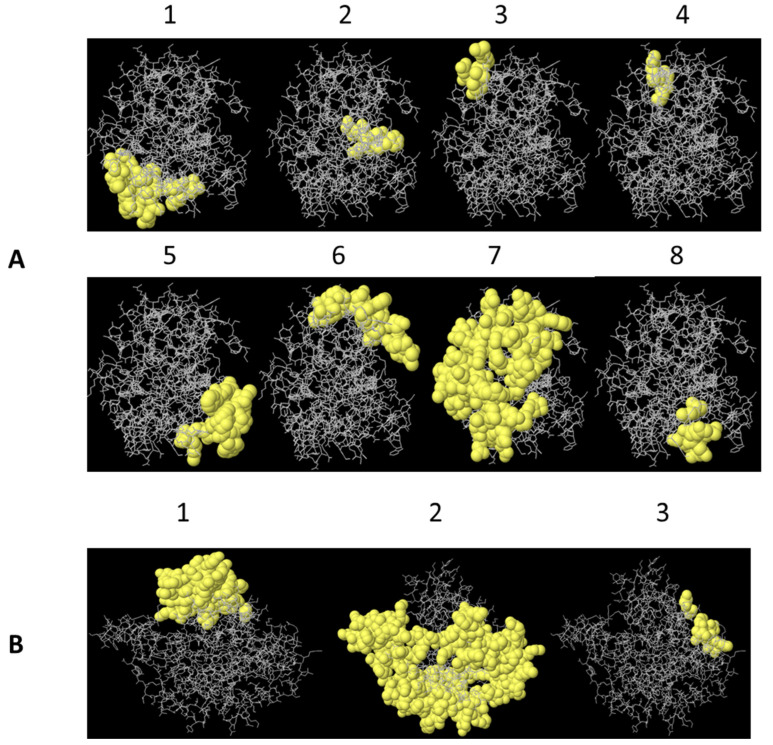
A schematic display of B lymphocyte discontinuous epitopes of PLD 2J-L (**A**) and CP40 2J-L (**B**) estimated by ElliPro program, based on the 3D structure of the proteins. Numbers represent each epitope in Table 3.

**Table 1 ijms-25-00270-t001:** B lymphocyte epitopes of PLD 2J-L and CP40 2J-L proteins predicted by the BepiPred 2.0 program.

Protein	No.	Epitopes	aa	Length	Score *
**PLD** **2J-L**	1	AAPVVHNPASTAN	24–36	13	0.511–0.639
	2	WWADHDGIPTSAGATAE	74–90	17	0.510–0.656
	3	DKRKQG	98–103	6	0.519–0.538
	4	DYCRDARSVC	117–126	10	0.515–0.552
	5	TVGGP	153–157	5	0.509–0.527
	6	QDVLNDFARSENKILTKQK	179–197	19	0.502–0.582
	7	YYNINQGFGNCYGTWNRTCDQLRKSSEARDQGKLG	203–237	34	0.505–0.622
	8	ATGQDAR	245–251	7	0.501–0.549
	9	THFYRHADTE	271–280	10	0.507–0.591
	10	DKHSATHHLATVA	291–303	13	0.518–0.551
**CP40** **2J-L**	1	PRSVS	5–9	5	0.498–0.526
	2	AESATLSKEPLKASPGRADTVGVQTTCN	31–58	28	0.508–0.670
	3	RDKAIQLKDDDPWKDKLQVKLTD	71–93	23	0.508–0.644
	4	DNQKSDQQFWETFHRE	108–123	16	0.532–0.594
	5	LLLNKIKDKNLYGKHVEDDYK	143–163	21	0.508–0.634
	6	ELRQVEKQLNLKWQ	189–202	14	0.501–0.644
	7	LMGPKAPANEGKKPDHEG	213–230	18	0.506–0.638
	8	DNAQTSQ	240–246	7	0.504–0.552
	9	DTKESVTQVWNGFRDKINSCQF	265–286	22	0.510–0.589
	10	PEENDTNRFLTAVGEVNKSGAMQVAEWKPEGGE	293–325	33	0.502–0.640
	11	RTYDGDDFTTLKPTDFAF	339–356	18	0.501–0.620
	12	ELTTGESSTDLGKPT	362–376	15	0.515–0.639

* Minimum and maximum values within the epitope’s entire sequence.

**Table 2 ijms-25-00270-t002:** B lymphocyte linear epitopes of PLD 2J-L and CP40 2J-L proteins determined by the ElliPro program.

Proteins	No.	aa Residues	Epitopes	Length	Score
**PLD** **2J-L**	1	9–33	MREKVVLFLSIIMAIMLPVGNAAAAPVVHNPAS	33	0.856
	2	244–250	IATGQDA	7	0.721
	3	78–91	HDGIPTSAGATAEE	14	0.698
	4	211–226	GNCYGTWNRTCDQLRK	16	0.689
	5	275–285	RHADTENSFKA	11	0.684
	6	175–201	SGPAQDVLNDFARSENKILTKQKIADY	27	0.656
	7	132–142	RDLARKYLEPA	11	0.655
	8	116–125	PDYCRDARSV	10	0.642
	9	47–50	LTTQ	4	0.613
	10	69–76	AWGRGWWA	8	0.612
	11	160–168	KTITADLRD	9	0.583
	12	229–236	EARDQGKL	8	0.550
	13	289–292	WVDK	4	0.535
**CP40** **2J-L**	1	151–166	KNLYGKHVEDDYKYRE	16	0.807
	2	220–229	ANEGKKPDHE	10	0.797
	3	9–46	MHNSPRSVSRLITVGITSALFASTFSAVSASATLSKEPLKASPG	46	0.749
	4	362–379	ELTTGESSTDLGKPTGSR	18	0.719
	5	337–355	DGRTYDGDDFTTLKPTDFA	19	0.707
	6	193–205	VEKQLNLKWQLRK	13	0.702
	7	294–315	EENDTNRFLTAVGEVNKSGAMQ	22	0.660
	8	262–281	YKKDTKESVTQVWNGFRDKI	20	0.638
	9	48–56	ADTVGVQTT	9	0.613
	10	318–326	EWKPEGGEK	9	0.604
	11	72–89	DKAIQLKDDDPWKDKLQV	18	0.588
	12	145–149	LNKIK	5	0.544
	13	240–245	DNAQTS	6	0.531

**Table 3 ijms-25-00270-t003:** B lymphocyte conformational epitopes of PLD 2J-L and CP40 2J-L proteins determined by the ElliPro program.

Protein	No.	Epitopes	Lengh	Score
**PLD** **2J-L**	1	_:M1, _:K4, _:V5, _:V6, _:L7, _:F8, _:L9, _:S10, _:I11, _:I12, _:M13, _:A14, _:I15, _:M16, _:L17, _:P18, _:V19, _:G20, _:N21, _:A22, _:A23, _:A24, _:A25, _:P26, _:V27, _:V28, _:H29, _:N30, _:P31, _:A32, _:S33, _:V290, _:D291, _:K292	34	0.822
	2	_:H276, _:A277, _:D278, _:T279, _:E280, _:N281, _:S282, _:F283	8	0.796
	3	_:A135, _:R136, _:L139, _:E140, _:P141, _:A142	6	0.726
	4	_:K101, _:Q102, _:G103	3	0.686
	5	_:F210, _:G211, _:N212, _:C213, _:Y214, _:G215, _:T216, _:W217, _:N218, _:R219, _:T243, _:I244, _:A245, _:T246, _:G247, _:Q248, _:D249, _:A250, _:D254, _:G257, _:K258	21	0.675
	6	_:L47, _:T48, _:T49, _:Q50, _:W75, _:A76, _:D77, _:D79, _:G80, _:I81, _:P82, _:T83, _:S84, _:A85, _:G86, _:A87, _:T88, _:E90, _:E91, _:D133, _:K137, _:Y138	22	0.662
	7	_:A69, _:W70, _:G71, _:R72, _:G73, _:W74, _:N115, _:P116, _:D117, _:Y118, _:R120, _:D121, _:A122, _:S124, _:V125, _:I128, _:N129, _:R132, _:K152, _:T153, _:V154, _:G155, _:G156, _:P157, _:K160, _:T161, _:I162, _:T163, _:A164, _:D165, _:L166, _:R167, _:D168, _:S175, _:G176, _:P177, _:A178, _:Q179, _:D180, _:L182, _:N183, _:D184, _:A186, _:R187, _:S188, _:E189, _:N190, _:K191, _:I192, _:L193, _:T194, _:K195, _:Q196, _:K197, _:I198, _:A199, _:D200, _:Y201, _:G202, _:Y203, _:E229, _:A230, _:D232, _:Q233, _:G234, _:K235, _:L236	67	0.609
	8	_:Q208, _:T220, _:C221, _:D222, _:Q223, _:R225, _:A259	7	0.529
**CP40** **2J-L**	1	_:L145, _:N146, _:K147, _:I148, _:K149, _:D150, _:K151, _:N152, _:L153, _:Y154, _:G155, _:K156, _:H157, _:V158, _:E159, _:D160, _:D161, _:Y162, _:K163, _:Y164, _:R165, _:E166, _:R169, _:V193, _:E194, _:K195, _:Q196, _:L197, _:N198, _:L199, _:K200, _:W201, _:Q202, _:R204, _:K205, _:S245	36	0.721
	2	_:M1, _:H2, _:N3, _:S4, _:P5, _:R6, _:S7, _:V8, _:S9, _:R10, _:L11, _:I12, _:T13, _:V14, _:G15, _:I16, _:T17, _:S18, _:A19, _:L20, _:F21, _:A22, _:S23, _:T24, _:F25, _:S26, _:A27, _:V28, _:A29, _:A31, _:E32, _:S33, _:A34, _:T35, _:L36, _:S37, _:K38, _:E39, _:P40, _:L41, _:K42, _:A43, _:S44, _:P45, _:G46, _:R47, _:A48, _:D49, _:T50, _:V51, _:G52, _:V53, _:Q54, _:T55, _:T56, _:C57, _:D72, _:K73, _:A74, _:I75, _:Q76, _:L77, _:K78, _:D79, _:D80, _:D81, _:P82, _:W83, _:K84, _:D85, _:K86, _:L87, _:Q88, _:V89, _:Q110, _:K111, _:D113, _:Q114, _:R122, _:A220, _:N221, _:E222, _:G223, _:K224, _:K225, _:P226, _:D227, _:H228, _:E229, _:F239, _:D240, _:N241, _:A242, _:Q243, _:K263, _:K264, _:D265, _:T266, _:K267, _:E268, _:S269, _:V270, _:T271, _:Q272, _:V273, _:N282, _:P293, _:E294, _:E295, _:N296, _:D297, _:T298, _:N299, _:R300, _:F301, _:L302, _:T303, _:A304, _:V305, _:G306, _:E307, _:V308, _:N309, _:K310, _:S311, _:G312, _:M314, _:Q315, _:E318, _:W319, _:K320, _:P321, _:E322, _:G323, _:G324, _:E325, _:D337, _:G338, _:R339, _:T340, _:Y341, _:D342, _:G343, _:D344, _:D345, _:F346, _:T347, _:T348, _:L349, _:K350, _:P351, _:T352, _:D353, _:F354, _:A355, _:R359, _:T365, _:G366, _:E367, _:S368, _:S369, _:T370, _:D371, _:L372, _:G373, _:K374, _:P375, _:T376, _:G377, _:S378, _:R379	171	0.683
	3	_:G248, _:L249, _:N275, _:G276, _:R278, _:D279, _:K280, _:I281	8	0.634

**Table 4 ijms-25-00270-t004:** Mapping of amino acid residues of the PLD 2J-L and CP40 2J-L proteins predicted as part of B lymphocyte epitopes identified by different prediction programs.

PLD 2J-L aa	BepiPred 2.0	Emini Surface Accessibility Scale	Antigenicity Kolaskar—Tongaonkar Scale	ElliPro Lineal	ElliPro Conformational
1–50	24 *AAPVVHNPASTAN* 36	31 *PASTANRP* 38	4 *KVVLFLSIIMAIMLPVGNA* 22 24AAPVVHNP 31 36NRPVYAIAHRVLTTQ 50	1 *MREKVVLFLSIIMAIMLPVGNAAAAPVVHNPAS* 33 47 LTTQ 50	1 *MKVVLFLSIIMAIMPVGNA* *AAAPVVHNPASVDK* 292 47 *LTTQWADDGIPTSAGATEE*DKY 138
51–130	74 *WWADHDGIPTSAG* *ATAE* 90 98 *DKRKQG* 103 117 *DYCRDARSVC* 126	96 IA*DKRKQ* 102 113 IK*NPDYCRDA* 122	53 DDAVAIGANALEID 66 116 *PDYCRDARSVCSIN* 129	69 AWGRGWWA 75 78 *HDGIPTSAGATAEE* 91 116 *PDYCRDARSV* 125	69 *AWGRGW*N*PDYRDASVINRKTVGGPKTITADLRDSGPAQDLNDARSEN* *KIL*TK*QKIADY*GY*EADQGKL* 236 101 *KQG* 103
131–160	153 *TVGGP* 157	133 *DLARKYL* 139	131 L*RDLARKYLEPA*GVRVLYGFYK*TVGGP* 157	132 *RDLARKYLEPA* 142	135 *ARLEPA* 142
161–240	179 *QDVLNDFARSENKILTKQK* 197 203 YYNINQGFGNCYGTWNRTCDQLRKSSEARDQGKLG 237	185 *FARSENKILTKQKI* 198 223 QLRKSSEARDQGKLG 237	171 AVAL*SGPAQDVLND* 184 198 IADYGYY 204	160 *KTITADLRD* 168 175 SGPAQDVLNDFARSENKILTKQKIADY 201 211 GNCYGTWNRTCDQLRK 226 229 EARDQGKL 236	208 QTCDQRA 259 210 FGNCYGTWNRT*IATGQDADGK* 258
240–260	245 *ATGQDA*R 251	NP	252 VNDLLGKA 259	244 I*ATGQDA* 250	*
261–290	271 *THFYRHADTE* 280	273 *YRHADTENS* 282	261 VDGLIFGFKI*THFYRH* 276	275 *RHADTENSFKA* 285	276 *HADTENSF* 283
291–307	291 *DKHSATHHLATVA* 303	289 *WVDKHSA* 295	295 *ATHHLATVA* 303	289 *WVDK* 292	*
**CP40 2J-L** **aa**	**BepiPred 2.0**	**Emini** **Surface** ** Accessibility Scale**	**Antigenicity** **Kolaskar—Tongaonkar scale**	**ElliPro (Lineal)**	**ElliPro (Discontinuous)**
1–70	5 PRSVS 9 31 AESATLSKEPLKAS PGRADTVGVQTTCN 58	NP	5 *PRSVSRLITVGITSALFASTFSAVASA* 31 33 *SATLSKEPLKAS* 44 50 *TVGVQTTCN*AKPIFFGY 66	1 MHNS*PRSVSRLITVGIT* *SALFASTFSAVASAESATLSKE* *PLKAS*PG 46	1MHNS*PRSVSRLITVGITSALFASTFSAAAESATLSKEPLKA* *S*PGR*ADTVGVQTTCDKAIQLKDDDPWKDKLQVQKDQRANEGKKPDHEFDNAQKKDTKESVTQVNPEENDTNRFLTA-VGEVNKSGMQEKPEGGE* *DGRTYD-GDDFTTLKPTDFART-GESSTDLGKPTGSR* 379
71–170	71 R*DKAIQLKDDDPWKDKLQVKLTD* 93 108 *DNQKSD*QQFWET*FHR*E 123 143 *LLLNKIKDKNLYGKHVEDDYK* 163	78 *KDDDPWKDKL* 87 107 E*DNQKSD* 113 158 VEDDYKYREI 167	86 *KLQVKLTD*IPEHVNMVSLFHVE 107 133 TRVVRTVGAQ*LLLNK* 147	72 *DKAIQLKDDDPWKDKLQV* 89 145 LNKIK 149 151 *KNLYGKHVEDDYKYRE* 166	145 *LNKIKDKNLYGKHVEDDYKYRE*R*VEKQLNLK*WQRKS 245
171–260	189 *ELRQVEKQLNLKWQ* 202 213 LMGPKAPANEGKKPDHEG 230 240 *DNAQTSQ* 246	221 *NEGKKP* *DHEG*Y 231	172 YNEYVVKHNL 181 189 *ELRQVEKQLNLK* 200 231 YKYLIYD 237 244 TSQVGLVADLVDYVLAQT 261	193 *VEKQLNLKWQ*LRK 205 220 *ANEGKKPDHE* 229 240 *DNAQTS* 245	*
261–325	265 *DTKESVTQVWNGFRDKINSCQF* 286 293 *PEENDTNRFLTAVGEVNKSGAMQ*VA*EWKPEGGE* 325	260 QT*YKKDTKES* 269 292 H*PEENDT* 298	281 *INSCQF*MAGYA 291 302 *LTAVGEV* 308	262 *YKKDTKESVTQVWNGFRDKI* 281 294 *EENDTNRFLTAVGEVNKSGAMQ* 315 318 *EWKPEGGE*K 326	248 *GL*N*GRDKI* 281 *
326–379	339 *RTYDGDDFTTLKPTDFA*F 356 362 ELTTGESSTDLGKPT 376	336 R*DGRTYDG* 343	337 *DGRTYDGDDFTTLKPTDFA* 355 362 *ELTTGESSTDLGKPTGSR* 379		*

NP: No prediction in that region of the sequence. The regions in the protein sequence that are identified as epitopes by different prediction programs are represented in cursive. *: Asterisks represent regions of the protein sequence with amino acid residues included in conformational epitopes described above in the table.

**Table 5 ijms-25-00270-t005:** T lymphocyte epitopes presented in the context of MHC class I, of the PLD 2J-L and CP40 2J-L proteins, and the aa residues that overlap as B cell epitopes.

Protein PLD 2J-L										
No	Allele	Start	End	Epitope	Score	BepiPred 2.0	Emini Surface Accessibility Scale	Antigenicity Kolaskar Tongaonkar Scale	Ellipro Lineal	Ellipro Conformational
1	H-2-Kb	146	154	VLYGFYKTV	0.916	153′TV′154	NP	146′VLYGFYKTV′154	NP	152′KTV′154
2	H-2-Kk	279	286	TENSFKAI	0.87	279′TE′280	279′TEN′281	NP	279′TENSFKA′285	279′TENSF′283
3	H-2-Kb	104	111	ANITFTWL	0.822	NP	NP	NP	NP	NP
4	H-2-Kk	89	96	AEEIFKHI	0.791	NP	96′I	NP	89′AEE′91	290′VDK′292
5	H-2-Kd	39	47	VYAIAHRVL	0.77	NP	NP	39′VYAIAHRVL′47	47′L	47′L
6	H-2-Db	56	65	VAIGANALEI	0.715	NP	NP	56′VAIGANALEI′65	NP	NP
7	H-2-Kk	139	147	LEPAGVRVL	0.715	NP	NP	139′LEPAGVRVL′147	139′LEPA′142	139′LEPA′142
8	H-2-Kd	202	210	GYYNINQGF	0.628	203′YYNINQGF′210	202′GYYNI′206	202′GYY′204	NP	202′GY203′208′Q 210′F
9	H-2-Kb	4	11	KVVLFLSI	0.62	NP	NP	4′KVVLFLSI′11	4′KVVLFLSI′11	4′KVVLFLSI′11
10	H-2-Kd	137	146	KYLEPAGVRV	0.609	NP	137′KYL′139	137′KYLEPAGVRV′146	137′KYLEPA′142	137′KYLEPA′142
11	H-2-Kb	199	206	ADYGYYNI	0.553	203′YYNI′206	201′YGYYNI′206	199′ADYGYY′204	199′ADY′201	199′ADYGY′203
12	H-2-Kk	139	146	LEPAGVRV	0.545	NP	139′L	139′LEPAGVRV′146	139′LEPA′142	139′LEPA′142
13	H-2-Kk	63	70	LEIDFTAW	0.526	NP	NP	63′LEID′70	69′AW′70	69′AW′70
14	H-2-Kb	56	63	VAIGANAL	0.524	NP	NP	56′VAIGANAL′63	NP	NP
15	H-2-Kb	266	273	FGFKITHF	0.522	271′THF′273	273′F	266′FGFKIT -HF′273	NP	NP
16	H-2-Qa1	138	146	YLEPAGVRV	0.516	NP	138′YL′139	138′YLEPAGVRV′146	138′YLEPA′142	139′LEPA′142
**Protein CP40 2J-L**										
1	H-2-Kk	173	181	NEYVVKHNL	0.943	NP	NP	173′NEYVVKHNL′181	NP	NP
2	H-2-Kk	294	302	EENDTNRFL	0.927	194′EENDTNRFL′202	194′EENDT′298	302′L	194′EENDTNRFL′202	194′EENDTNRFL′202
3	H-2-Kk	158	167	VEDDYKYREI	0.694	158′VEDDYK′163	158′VEDDYKYREI′167	NP	158′VEDDYKYRE′166	158′VEDDYKYRE′166
4	H-2-Kb	205	213	KIMGAFSEL	0.672	NP	NP	NP	205′K	NP
5	H-2-Qa1	137	145	RTVGAQLLL	0.664	143′LL′145	NP	137′RTVGAQLLL′145	145′L	145′L
6	H-2-Kb	206	213	IMGAFSEL	0.654	213′L	NP	NP	NP	NP
7	H-2-Kk	228	235	HEGYKYLI	0.651	228′HEG′230	228′HEGY′231	231′YKYLI′235	228′HE′229	228′HE′229,
8	H-2-Qa1	333	341	ALDRDGRTY	0.645	339′RTY′341	336′RDGRTY′341	NP	337′DGRTY′341	339′R
9	H-2-Kk	172	181	YNEYVVKHNL	0.61	NP	NP	172′YNEYVVKHNL′181	NP	NP
10	H-2-Kd	163	171	KYREIARDV	0.58	163′K	163′KYREI′167	NP	163′KYRE′166	163′KYRE′166 169′R 171′V
11	H-2-Kk	294	301	EENDTNRF	0.568	294′EENDTNRF′301	294′EENDT′298	NP	294′EENDTNRF′301	294′EENDTNRF′301
12	H-2-Qa1	205	213	KIMGAFSEL	0.539	213′L	NP	NP	205′K	NP
13	H-2-Kb	339	349	RTYDGDDFTTL	0.53	339′RTYDGDDFTTL′349	339′MRTYDG’ 343	NP	339′RTYDGDDFTTL′349	339′RTYDGDDFTTL′349
14	H-2-Kb	135	143	VVRTVGAQL	0.52	143′L	NP	135′VVRTVGAQL′143	NP	NP
15	H-2-Kb	17	25	TSALFASTF	0.502	NP	NP	17′TSALFASTF′25	17′TSALFASTF′25	17′TSALFASTF′25

**Table 6 ijms-25-00270-t006:** T lymphocyte epitopes presented in the context of MHC class II, of the PLD 2J-L and CP40 2J-L proteins and the aa residues that overlap as B cell epitopes.

	Protein PLD 2J-L									
No	Allele	Start	End	Epitope	Score	BepiPred 2.0	Emini Surface Accessibility Scale	Antigenicity Kolaskar Tongaonkar Scale	Ellipro Lineal	Ellipro Conformational
1	H2-IAb	24	38	AAPVVHNPASTANRP	0.6864	24′AAPVVHNPASTAN′36	31′PASTANRP′38	24′AAPVVHNP′31, 36′NRP′38	24′AAPVVHNPAS′33	24′AAPVVHNPAS′33
2	H2-IAb	25	39	APVVHNPASTANRPV	0.6886	25′APVVHNPASTAN′36	31′PASTANRP′38	36′NRPV′39	25′APVVHNPAS′33	25′APVVHNPAS′33
3	H2-IAb	20	34	GNAAAAPVVHNPAST	0.662	24′AAPVVHNPAST′34	31′PAST′34	24′AAPVVHNP′31	20′GNAAAAPVVHNPAS′33	20′GNAAAAPVVHNPAS′33
4	H2-IAb	271	285	THFYRHADTENSFKA	0.6544	271′THFYRHADTE′280	273′YRHADTENS′282	271′THFYRH′276	275′RHADTENSFKA′285	276′HADTENSF′283
5	H2-IAb	19	33	VGNAAAAPVVHNPAS	0.6458	24′AAPVVHNPAS′33	31′PAS′33	24′AAPVVHNP′31	19′VGNAAAAPVVHNPAS′33	19′VGNAAAAPVVHNPAS′33
6	H2-IAb	270	284	ITHFYRHADTENSFK	0.6179	271′THFYRHADTE′280	273′YRHADTENS′282	270′ITHFYRH′276	275′RHADTENSFK′284	276′HADTENSF′283
7	H2-IAb	18	32	PVGNAAAAPVVHNPA	0.546	24′AAPVVHNPA′32	31′PA′32	24′AAPVVHNP′31	20′PVGNAAAAPVVHNPA′32	18′PVGNAAAAPVVHNPA′32
8	H2-IAb	23	37	AAAPVVHNPASTANR	0.5471	24′AAPVVHNPASTAN′36	31′PASTANR′37	24′AAPVVHNP′31, 36′NR′37	23′AAAPVVHNPAS′33	23′AAAPVVHNPAS′33
9	H2-IAd	167	181	RDGEAVALSGPAQDV	0.6131	179’ QDV’ 181	NP	171 AVALSGPAQDV′181	167′RD′168, 175′SGPAQDV′181	167′RD′168, 175′SGPAQD′180
10	H2-IAd	166	180	LRDGEAVALSGPAQD	0.5894	179’ QD’ 180	NP	171 AVALSGPAQD′180	166′DLRD′168	166′LRD′168, 175′SGPAQD′180
**Protein CP40 2J-L**										
1	H2-IAb	210	224	FSELMGPKAPANEGK	0.7198	213′LMGPKAPANEGK′224	221′NEGK′224	NP	220′ANEGK′224	220′ANEGK′224
2	H2-IAb	209	223	AFSELMGPKAPANEG	0.6642	213′LMGPKAPANEG′223	221′NEG′223	NP	220′ANEG′223	220′ANEG′223
3	H2-IAb	208	222	GAFSELMGPKAPANE	0.5786	213′LMGPKAPANEG′222	221′NE′222	NP	220′ANE′222	220′ANE′222
4	H2-IAb	286	300	FMAGYAHPEENDTNR	0.5793	286′F, 293′PEENDTNR′300	292 HPEENDT 298	286′FMAGYA′291	294′EENDTNR′300	293′PEENDTNR′300
5	H2-IAb	211	225	SELMGPKAPANEGKK	0.5847	213′LMGPKAPANEGKK 225	221′NEGKK′225	NP	220′ANEGK′225	220′ANEGK′224
6	H2-IAb	283	297	SCQFMAGYAHPEEND	0.5853	283′SCQF′286, 293′PEEND′297	292′HPEEND’ 297	283′NSCQFMAGYA′291	294′EEND′297	293′PEEND′297
7	H2-IAb	287	301	MAGYAHPEENDTNRF	0.5647	293′PEENDTNRF′301	292 HPEENDT 298	287′FMAGYA′291	294′EENDTNRF′301	293′PEENDTNRF′301
8	H2-IAd	243	257	QTSQVGLVADLVDYV	0.6316	243′QTSQ′246	NP	244′TSQVGLVADLVDY′256	243′QTS′245	243′Q, 245′S, 248′GL′249
9	H2-IAd	242	256	AQTSQVGLVADLVDY	0.6365	243′AQTSQ′246	NP	244′TSQVGLVADLVDYV′258	243′QTS′245	242′AQ′243, 245′S, 248′GL′249
10	H2-IAb	285	299	QFMAGYAHPEENDTN	0.5379	285′QF′286, 293′PEENDTN′325	292′HPEENDT′298	285′QFMAGYA′291	294′EENDTN′299	293′PEENDTN′299
11	H2-IAd	265	279	DTKESVTQVWNGFRD	0.612	265′DTKESVTQVWNGFRD′279	265′DTKES′269	NP	265′DTKESVTQVWNGFRD′279	265′DTKESVTQV′273
12	H2-IAb	285	299	QFMAGYAHPEENDTN	0.5379	285′QF′286, 293′PEENDTN′299	292′HPEENDT′298	285′QFMAGYA′291	294′EENDTN′299	293′PEENDTN′299
13	H2-IAd	265	279	DTKESVTQVWNGFRD	0.612	265′DTKESVTQVWNGFRD′279	265′DTKES′269	NP	265′DTKESVTQVWNGFRD′279	265′DTKESVTQV′273

## Data Availability

Not applicable.

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
