# Peer review of "Bioinformatic Approach of B and T Cell Epitopes of PLD and CP40 Proteins of Corynebacterium pseudotuberculosis ovis Mexican Isolate 2J-L towards a Peptide-Based Vaccine"

_ijms, 2023, doi:10.3390/ijms25010270_

Round 1

Reviewer 1 Report

Comments and Suggestions for Authors

The topic of the manuscript is of interest. Authors performed comprehensive analysis to come up with significant conclusions. However, there are some issues with this manuscript, mainly the discussion section and arranging the format of tables. The manuscript should be improved carefully. I came up with some corrections and questions. I will explain my viewpoints in more detail below.

1.       Table 4, 5 and 6 need to be arranged in proper format, the current table size is very large. For example, all alleles and peptides are in two lines and occupy extra spaces. It can be in small font size or increase the width of the tables.

2.       Table 3 needs to be designed in proper format with the separate rows showing lines of Epitopes numbers.

3.       Page 9, line 177, it says “by different prediction programs are represented in bold.” However, it could be in different ways such as programs can be defined by a symbol or a number in superscript on the header of the table (e.g., Bepidred 2.01, Emini Surface Accessibility Scale2 and these numbers can be used in the table where epitopes were identified by different prediction programs (e.g., WWADHDGIPTSAGATAE1,2).

4.       Several tools and software are described in the manuscript without the citations especially in “Materials and Methods”. For example, “Phyre2” line 495, “HHpred 1.51”, “Psi-pred 2.5” line 498, “Jmol viewer” line 501, “Immune epitope database and resource website” line 503. Please check the whole manuscript carefully to cite all the software and tools.

5.       Page 16, line 320, (Barral et al., 2022) not in format.

6.       Re-write discussion part, reduce the size, and avoid describing the software and tools, write directly about the findings and correlate the findings with the supporting studies.

Several paragraphs in the discussion describe the tools and software that look like their definitions. For example, page 15, line 230 to 234, the paragraph describes the IEDB. This information should be removed from the discussion.

Another line 255 to 264, it sounds the paragraph is defining more about the B cell epitopes than discussing the findings in the manuscript. Similarly, it should be re-considered writing from line 265 to 288 page 15 and page 16.

7.       Again lines 301 to 305, the paragraph is describing the “ElliPro program”, not discussing the finding in the manuscript. It should be removed or improved.

8.       Line 295, it says “the different programs used”, which programs?

Author Response

The authors appreciate the review carried out on the manuscript and have worked to modify the document following the reviewer's recommendations.

  1. Table 4, 5 and 6 need to be arranged in proper format, the current table size is very large. For example, all alleles and peptides are in two lines and occupy extra spaces. It can be in small font size or increase the width of the tables.Authors response: Tables were corrected to the journal format, and the font was reduced.
  2. Table 3 needs to be designed in proper format with the separate rows showing lines of Epitopes numbers. Author's response: The Table 3 was corrected.
  3. Page 9, line 177, it says “by different prediction programs are represented in bold.” However, it could be in different ways such as programs can be defined by a symbol or a number in superscript on the header of the table (e.g., Bepidred 2.01, Emini Surface Accessibility Scaleand these numbers can be used in the table where epitopes were identified by different prediction programs (e.g., WWADHDGIPTSAGATAE1,2). Author's response: Table 4 aims to inform about the amino acid residues of the proteins that were identified as part of epitopes according to the different programs. Therefore, a mapping is carried out through the sequence, and the regions that were involved as part of an epitope are specified. The columns already report each predictive program and which regions of the sequence were estimated as epitopes by these programs. The objective of indicating in bold is to point out which amino acid residues coincide as part of epitopes in the different programs. We reduce the table, eliminating the regions where no epitopes were found. We are still modifying the table.
  4. Several tools and software are described in the manuscript without the citations especially in “Materials and Methods”. For example, “Phyre2” line 495, “HHpred 1.51”, “Psi-pred 2.5” line 498, “Jmol viewer” line 501, “Immune epitope database and resource website” line 503. Please check the whole manuscript carefully to cite all the software and tools. Author's response: References for each program were added.       
  5. Page 16, line 320 (Barral et al., 2022) not in format. Author's response: The correct citation format was added.       
  6. Re-write discussion part, reduce the size, avoid describing the software and tools, write directly about the findings and correlate the findings with the supporting studies.

Several paragraphs in the discussion describe the tools and software that look like their definitions. For example, page 15, line 230 to 234, the paragraph describes the IEDB. This information should be removed from the discussion.

Another line 255 to 264, it sounds the paragraph is defining more about the B cell epitopes than discussing the findings in the manuscript. Similarly, it should be re-considered writing from line 265 to 288 page 15 and page 16.

6. Authors response: Modifications were made in the discussion section and were marked in yellow. We are still modifying these section.

  1. Again lines 301 to 305, the paragraph is describing the “ElliPro program”, not discussing the finding in the manuscript. It should be removed or improved. Author's response: Modifications were made in the discussion section and were marked in yellow. We are still modifying these section.
  2. Line 295, it says “the different programs used”, which programs?Author's response: The programs used were specified

Reviewer 2 Report

Comments and Suggestions for Authors

The manuscript ijms-2698234 entitled “Bioinformatic approach of B and T cell epitopes of PLD and CP40 proteins of Corynebacterium pseudotuberculosis ovis  Mexican isolate 2J-L, towards a peptide-based vaccine” reports the results of the bioinformatic strategy used to explore the existence of B and T cell epitopes of  Phospholipase D (PLD) and the Endoglycosidase CP40 of the veterinarian strain of C. pseudotuberculosis ovis 2J-L. The authors nicely in frame the context of the study. The results are properly analysed and the methodology is appropriate. Overall, the study shows scientific value.

Some English improvement is required.

The manuscript is recommended for publication.

Comments on the Quality of English Language

Author Response

The authors appreciate the review carried out on the manuscript and have worked to modify the document following the reviewer's recommendations.

Reviewer 3 Report

Comments and Suggestions for Authors

In this manuscript, the authors used computational methods to identify and analyze in silico potential B and T cell epitopes on two important virulence factors of an isolate of Corynebacterium pseudotuberculosis biovar ovis, the etiologic agent of CLA in small ruminants. Studies such as this one demonstrate the utility of in silico approaches in identifying potential peptide antigens for further immunological characterization in animal models. In this study, a number of peptide epitopes for B and T cell recognition were predicted for PLD and CP40. The authors need to discuss which peptide epitopes they would prioritize for in-depth functional/immunological evaluation. My comments below focus on the presentation of this research which requires substantial revision.

(1) The authors should specify that "ovis" is a biovar of C. pseudotuberculosis.

(2) The Introduction is too long and needs to be shortened to focus on the causative agent of CLA, the targeted virulence factors, and the murine model for experimental CLA vaccine evaluation. The background information on epitope mapping and reverse vaccinology should be substantially consolidated.

(3) Tables 4, 5, and 6 are almost unreadable because of formatting issues. The quality of presentation of the data in these tables needs to be extensively revised.

(4) The Discussion section should be extensively revised to remove redundancy and unnecessary background information. The authors restate a lot of background material in the Discussion that was already provided in the Introduction (e.g., IEDB website, linear and conformational epitopes). In addition, the background information on the different bioinformatic programs used is not necessary if these computational methods are established. Furthermore, the introductory information on basic immunology (e.g., MHC classes and the genetic polymorphism of HLA genes) should be removed. These are just examples, and the authors should carefully review their extremely long Discussion to identify other opportunities to remove or consolidate text.

Comments on the Quality of English Language

The quality of the English in this manuscript must be extensively revised to improve readability and clarity. 

Author Response

The authors appreciate the review carried out on the manuscript and have worked to modify the document following the reviewer's recommendations.

​The authors should specify that "ovis" is a biovar of  pseudotuberculosis.

1, Author's response: We specify that ovis is a biovar.

  • The Introduction is too long and needs to be shortened to focus on the causative agent of CLA, the targeted virulence factors, and the murine model for experimental CLA vaccine evaluation. The background information on epitope mapping and reverse vaccinology should be substantially consolidated.
  1. Author's response: We reduce the introduction, leaving only a paragraph for epitope mapping and reverse vaccinology.
  • Tables 4, 5, and 6 are almost unreadable because of formatting issues. The quality of presentation of the data in these tables needs to be extensively revised.
  1. Author's response: The tables were corrected
  • The Discussion section should be extensively revised to remove redundancy and unnecessary background information. The authors restate a lot of background material in the Discussion that was already provided in the Introduction (e.g., IEDB website, linear and conformational epitopes). In addition, the background information on the different bioinformatic programs used is not necessary if these computational methods are established. Furthermore, the introductory information on basic immunology (e.g., MHC classes and the genetic polymorphism of HLA genes) should be removed. These are just examples, and the authors should carefully review their extremely long Discussion to identify other opportunities to remove or consolidate text.
  1. Author's response: Modifications were made in the discussion section and were marked in yellow. We are still modifying these section.

Round 2

Reviewer 1 Report

Comments and Suggestions for Authors

The manuscript has been improved significantly, however, Table 4 can be improved to better represent it.

Author Response

A revision was made following the reviewers' suggestions and the changes are marked in yellow.

Reviewer 3 Report

Comments and Suggestions for Authors

Authors have responded adequately to my suggestions for revision and have improved the publication quality of their manuscript.Some minor issues with the English language remain.

Comments on the Quality of English Language

Some minor issues with the English language remain.

Author Response

(The authors gave the same response as above.)
